# Effects of Sodium Alginate Infusion on Intramammary Immunity Against Subclinical Mastitis in Dairy Cows

**DOI:** 10.3390/ijms26125515

**Published:** 2025-06-09

**Authors:** Yu-I Pan, Yu-Chia Lin, Jai-Wei Lee, Perng-Chih Shen, Rolissa Ballantyne, Hsu-Hsun Lee, Kuo-Hua Lee

**Affiliations:** 1Department of Veterinary Medicine, College of Veterinary Medicine, National Pingtung University of Science and Technology, Pingtung 912301, Taiwan; cornyui@gmail.com; 2Veterinary Medical Teaching Hospital, Department of Veterinary Medicine, College of Veterinary Medicine, National Pingtung University of Science and Technology, Pingtung 912301, Taiwan; 3Research Center of Animal Biologics, National Pingtung University of Science and Technology, Pingtung 912301, Taiwan; 4Department of Animal Science, National Pingtung University of Science and Technology, Pingtung 912301, Taiwan; jessica95089344@gmail.com (Y.-C.L.); pcshen@mail.npust.edu.tw (P.-C.S.); 5Department of Tropical Agriculture and International Cooperation, National Pingtung University of Science and Technology, Pingtung 912301, Taiwan; joeylee@mail.npust.edu.tw (J.-W.L.); rolissa12@gmail.com (R.B.); 6Northern Region Branch, Taiwan Livestock Research Institute, Ministry of Agriculture, Miaoli 368003, Taiwan

**Keywords:** dairy cows, immune modulation, sodium alginate, somatic cell count, subclinical mastitis

## Abstract

Mastitis is a major issue in dairy cows, with subclinical mastitis (SCM) being hard to detect and potentially progressing to clinical mastitis. Antibiotic use raises concerns about resistance and milk contamination, highlighting the need for natural alternatives. Sodium alginate (SA), known for its antioxidant and immunomodulatory properties, may offer a solution, though its effects on mastitis are unclear. Intramammary infusion of 1% SA (30 mL) was tested in both healthy cows (*n* = 8; somatic cell count, SCC ≤ 100,000 cells/mL) and those with SCM (*n* = 12; SCC ≥ 200,000 cells/mL). The results showed that SA significantly increased SCC in both healthy and SCM cows, with peak levels at 48 h, returning to baseline levels thereafter. In cows with SCM, SA treatment led to a 58.3% cytological and 54.5% bacteriological cure rate after 14 days. Additionally, significant downregulation was observed in tumor necrosis factor (TNF)-α, interleukin (IL)-1β, IL-2, IL-4, IL-6, and interferon (IFN)-γ. Conversely, the levels of IL-8, IL-10, and IL-12 initially increased, then declined gradually. Importantly, there were no significant effects on milk composition. These findings suggest that SA may offer an alternative to antibiotics, aiding in immune response and bacterial clearance without the risk of antibiotic residues, thus preventing SCM progression to clinical mastitis.

## 1. Introduction

Bovine mastitis is one of the most significant diseases affecting dairy farms worldwide due to its high prevalence and substantial economic impact. The global prevalence of clinical and subclinical mastitis has been reported to range from 26% to 63% [1,2,3,4]. This disease leads to considerable financial losses due to reduced milk quality and quantity, increased veterinary expenses, and the premature culling of affected cows [5,6]. Bovine mastitis is an inflammatory condition of the udder tissue caused by the invasion of microorganisms which increases the somatic cell count (SCC) in milk [7]. The disease can be classified into two forms: clinical and subclinical. Clinical mastitis is characterized by the presence of visible symptoms, while subclinical mastitis (SCM) is asymptomatic, thereby posing greater challenges for detection and diagnosis. Despite the lack of visible signs, SCM significantly affects milk composition and quality [8]. Notably, the prevalence of SCM is 15 to 40 times higher than that of clinical mastitis [9], making it more threatening to dairy production.

In general, the immune system is activated to eliminate invading pathogens, primarily through innate immune responses, which serve as the first line of defense [10]. These responses act immediately to destroy invading pathogens and prevent them from establishing infections. In the mammary gland, the innate immune response is initiated when milk macrophages and mammary epithelial cells encounter pathogens and release pro-inflammatory cytokines, including tumor necrosis factor (TNF)-α, interleukin (IL)-1β, IL-6, and IL-8 [7,11]. These cytokines trigger the recruitment of leukocytes from the bloodstream to the site of infection. As a result, SCC in milk increases, mostly neutrophils, during mastitis, and commonly being used as an indicator of mastitis [7]. Upon infiltration, neutrophils phagocytose and destroy pathogens intracellularly which is facilitated by cytokines such as IL-12 and interferon (IFN)-γ [12]. To restore homeostasis, IL-4 and IL-10 play crucial roles in limiting leukocyte infiltration and promoting the termination of inflammation [11]. However, in the case of SCM, the immune system is not activated enough to eradicate pathogens. On the other hand, the pathogens are being suppressed by the immune system and not able to induce clinical symptoms, leaving SCM difficult to detect. As a consequence, cows with SCM are not treated and keep spreading pathogens in the herd.

Intramammary antibiotic infusion is the primary method for treating mastitis in most dairy farms worldwide [13]. However, excessive antibiotic use presents several challenges, including the emergence of antibiotic-resistant bacteria, food safety, and public health concerns [14]. Given these limitations, developing alternative non-antibiotic treatments has become a critical research focus to ensure effective mastitis control while minimizing adverse consequences. Sodium alginate (SA), a polysaccharide biopolymer derived from brown seaweed (Phaeophyceae), has attracted significant attention due to its broad applications in the food, agriculture, and biomedical industries. In addition to its well-documented biocompatibility and non-cytotoxicity [15], sodium alginate exhibits antioxidant [16], immunomodulatory [17], and immunostimulatory [18] properties, making it a promising candidate for veterinary and animal health applications. Recent studies have demonstrated that alginate supplementation enhances immune function in various species, including fish [19], poultry [20], pigs [21], and rats [22]. However, research on its effects on treating bovine mastitis remains unclear. Therefore, this study aims to investigate whether the intramammary infusion of sodium alginate is able to elicit immune responses in healthy quarters. Moreover, the therapeutic effects of intramammary sodium alginate infusion were investigated in quarters with SCM.

## 2. Results

In experiment 1, the SCC exhibited a significant increase on day 2 (D2) following SA infusion, after which it gradually declined (Figure 1a). However, no significant differences were observed in milk composition, including fat (*p* = 0.731), protein (*p* = 0.657), lactose (*p* = 0.644), and solids-not-fat (SNF) (*p* = 0.648), between D0 and D7 (Appendix A). Furthermore, the analysis of oxidative stress and immune cell function revealed no significant changes in the phagocytic activity (*p* = 0.972; Figure 2a) or reactive oxygen species (ROS) levels (*p* = 0.155; Figure 2b) of somatic cells before and after SA administration. Three cases were excluded from the ROS and phagocytic activity analysis due to insufficient sample volume.

The results of experiment 2 showed a similar trend as those of experiment 1. Following SA infusion, the somatic cell score (SCS) significantly increased at 12 h post-administration, reaching its peak at 48 h. Subsequently, SCS returned to baseline levels by D7 and continued to decline through D14 (Figure 1b). Despite these fluctuations in SCS, no significant differences were detected in milk composition parameters, including fat (*p* = 0.319), protein (*p* = 0.240), lactose (*p* = 0.147), and SNF (*p* = 0.384), before and after SA infusion (Appendix A).

Regarding treatment efficacy, the cytological cure rate at D14 was 58.3% (7 of 12), while the bacteriological cure rate was 54.5% (6 of 11) (Figure 3). One case was excluded from the bacteriological cure rate analysis due to milk culture contamination. Additionally, no significant changes were observed in phagocytic activity (*p* = 0.833; Figure 2c) or ROS levels (*p* = 0.225; Figure 2d) following SA infusion.

The expression levels of TNF-α (Figure 4a), IL-1β (Figure 4b), IL-6 (Figure 4c), IL-2 (Figure 5a), IFN-γ (Figure 5c), and IL-4 (Figure 6a) significantly decreased after infusion, reaching their lowest levels at 24 h post-administration. However, TNF-α (Figure 4a), IL-1β (Figure 4b), and IL-6 (Figure 4c) gradually returned to baseline from D2 to D7. In contrast, IL-2 (Figure 5a) and IL-4 (Figure 6a) rapidly returned to baseline by D2, whereas IFN-γ (Figure 5c) exhibited an increasing trend at 48 h post-SA treatment (*p* = 0.072). On the other hand, IL-8 (Figure 4d) and IL-10 (Figure 6b) expression levels significantly increased at 12 h post-administration, followed by a subsequent decline at 48 h before returning to baseline. Additionally, IL-12 (Figure 5b) expression was upregulated within 48 h post-SA treatment and then gradually decreased to baseline levels.

## 3. Discussion

In this study, a rapid increase in SCC was observed following SA infusion, suggesting that SA possesses immunostimulatory properties. Sodium alginate has been shown to enhance immune responses by stimulating the nuclear factor kappa-light-chain-enhancer of activated B cells (NF-κB) and mitogen-activated protein kinase (MAPK) signaling pathways, leading to the upregulation of several cytokine genes [18,24]. The accumulation of immune cells likely facilitated pathogen clearance by releasing ROS and engulfing invading pathogens. Consequently, the bacteriological cure rate reached 54.5% after 14 days of treatment. Notably, inflammation resolved quickly, with SCC returning to pre-treatment levels within 7 days and then declining continuously, reaching an even lower level by day 14. Previous studies have demonstrated that the cytological cure rate in cattle with clinical mastitis ranged from 18.7% to 22.3% when treated with conventional antibiotics [25,26]. Additionally, cure rates of 42% to 45% have been reported across various mastitis cases [27]. In our study, the cytological cure rate of SCM was found to be 58.3%. Although the bacteriological cure rate of sodium alginate was lower than that of conventional antibiotic therapy (73.3–75.3%) [25,26], this can be attributed to its lack of direct antibacterial properties. However, the high cytological cure rate and the continual reduction in SCC suggest that SA exerts immunomodulatory effects and enhanced innate immunity to cure SCM. More importantly, our findings indicate that SA infusion does not alter milk composition. Since the reduction in SCC is linked to improved milk production and increased economic returns in the dairy industry, these results highlight that SA could be a promising non-antibiotic alternative for treating SCM.

Previous research has indicated that alginate enhances phagocytic activity and ROS production in a dose-dependent manner in vitro [15,28]. However, in our study, SA infusion did not significantly alter phagocytic activity or ROS release levels in either healthy or SCM cows. This discrepancy may be due to the presence of milk components, such as fat globules and casein, which can interfere with the phagocytic capacity of polymorphonuclear neutrophils (PMNs) and contribute to immune cell exhaustion [29]. Additionally, previous studies have reported a negative correlation between mastitis severity and PMN phagocytic capacity [30,31], as well as reduced ROS production by milk PMNs due to continuous stimulation by milk components [32]. These findings suggest that while alginate may enhance immune function in vitro, its effects in vivo may be influenced by the complex milk environment and interactions among immune cells, potentially limiting its immunostimulatory effects in mastitis cows.

In addition, studies have also demonstrated that alginates can stimulate the production of TNF-α, IL-1β, and IL-6 in both human monocytes [33] and mice macrophages [34], highlighting the immunostimulatory effects of SA. The increased levels of TNF-α, IL-1β, and IL-6 promote endothelial activation and leukocyte recruitment to the site of infection, which plays a crucial role in the host’s innate immune defense [11]. In the present study, a significant increase in SCC was observed 12 h after SA treatment in both healthy and SCM cows. However, analysis of cytokine gene expressions was only conducted in experiment 2 to elucidate more detailed immunomodulating properties of SA in SCM cows. Despite the increase in SCC, TNF-α and IL-1β expression levels were significantly downregulated at 12 h compared to baseline (0 h), with TNF-α remaining low for up to seven days and IL-1β for two days. Similarly, IL-6 expression was downregulated between one and two days post-SA infusion. Notably, TNF-α, IL-1β, and IL-6 gene expression in lipopolysaccharide (LPS)-challenged mastitis models peaked at three hours, followed by a gradual decline, reaching the lowest levels at 12 h in liver and udder tissue cells [35,36]. This response appears to occur significantly earlier than the SCC peak (24 to 48 h) observed in another challenge study [37]. These findings suggest that the upregulation of these pro-inflammatory cytokines following SA administration is largely complete within 12 h in the current study. On the other hand, the relative gene expression of TNF-α and IL-1β in SCC is higher in SCM cows than in healthy cows, leading to an overall higher SCC in SCM cases [38]. The observed downregulation of these pro-inflammatory cytokines suggests that SA may exert a sustained anti-inflammatory effect following the initial accumulation of SCC, facilitating a more rapid reduction in SCC and ultimately contributing to a 58.3% cytological cure rate, with SCC levels decreasing below pre-treatment levels by day 14.

Interleukin-8 plays a crucial role in recruiting neutrophils to infection sites and enhancing their respiratory burst activity [39]. Previous studies have shown that IL-8 expression in milk SCC significantly increases at 16 h and 24 h following *Escherichia coli* and *Staphylococcus aureus* infections, respectively [37]. Interleukin-8 expression is induced not only by exogenous stimuli but also by endogenous pro-inflammatory factors such as TNF-α and IL-1β [11], leading to a delayed peak compared to other pro-inflammatory cytokines. In this study, IL-8 expression was upregulated 12 h after SA treatment, coinciding with an increase in milk SCC, further supporting the immunostimulatory effects of SA. Despite IL-8’s known capacity to enhance neutrophil respiratory burst activity, our findings did not detect a significant effect on this aspect.

Interferon-γ enhances macrophage-mediated pathogen killing, modulates neutrophil phagocytic activity, and primes cells for increased reactive oxygen species production [40,41]. Furthermore, IL-12 is a key inducer of IFN-γ production [42]. Yoshida et al. (2004) reported that co-culture with alginate oligosaccharides induces IFN-γ production in murine lymph node cells by stimulating IL-12 production while simultaneously suppressing IL-4 expression [43]. Similarly, another study demonstrated that intraperitoneal injection of alginate oligosaccharides significantly increased IFN-γ and IL-12 concentrations in mouse serum, whereas IL-4 remained undetectable. Consistent with these findings, our study also observed an upregulation of IL-12 following SA treatment, which subsequently led to a tendency for increased IFN-γ expression. These findings suggest that SA regulates immune responses by activating neutrophils and promoting their accumulation in udder tissue in dairy cows with mastitis. Additionally, the observed downregulation of IL-4 and IL-2 following SA infusion in SCM cows indicates that its immunomodulatory effects primarily enhance immune cell phagocytic activity rather than antibody production, which has a limited role in aiding mastitis recovery [7].

Interleukin-10 plays a key role in resolving inflammation by suppressing pro-inflammatory cytokines, chemokines, and eicosanoids in monocytes, macrophages, and neutrophils [44]. During mastitis infections, the initial production of IL-10 is caused by an increase in milk TNF-α concentrations, likely due to TNF-α’s ability to stimulate IL-10 production [11]. In this study, the increased expression of IL-10 at 12 h post-SA infusion suggests a preceding rise in TNF-α expression, supporting our earlier hypothesis. Subsequently, the downregulation of IL-10 may be attributed to the decline in TNF-α levels, indicating a tightly regulated inflammatory response.

## 4. Materials and Methods

### 4.1. Animals

The experiments were carried out at a commercial dairy farm in Pingtung County, Taiwan. In experiment 1, eight healthy lactating Holstein cows (*n* = 8) with an average parity of 2.0 ± 0.26 were selected. Their quarter milk samples were collected two days before the experiment for bacterial culture and SCC analysis to confirm the absence of mastitis (SCC less than 100,000 cells/mL and no bacterial growth). The average SCC of the samples was 72,750 ± 17,670 cells/mL. In experiment 2, twelve cows with quarters exhibiting SCC greater than 200,000 cells/mL and bacterial growth without visible symptoms (*n* = 12) were classified as cows with SCM. These cows had an average parity of 2.1 ± 0.34 and an average SCC of 541,417 ± 81,708 cells/mL. All experimental procedures were approved by the Institutional Animal Care and Use Committee (IACUC) of the National Pingtung University of Science and Technology (NPUST) (NPUST-113-098).

### 4.2. Preparation of SA for Intramammary Infusion (IMI)

The SA solution for IMI was prepared by resolving (1%) SA (Sigma-Aldrich, St. Louis, MO, USA) in deionized water with a homogenizer (Silverson L5M-A, Chesham, UK). Thereafter, 30 mL of SA solution was loaded into a syringe with a plastic cannula and sterilized by γ-radiation. The SA solution was plated on agar plates and incubated for 24 h to confirm its sterilization before being used for IMI.

### 4.3. Administration and Sampling

For both experiments, two syringes containing SA were administered via intramammary infusion at 12 h intervals following routine milking. D0, defined as the time of the first infusion, marked the start of sample collection. Quarter milk samples (100 mL) were collected aseptically on D0, D2, and D7 for experiment 1 and on D0, D0.5 (12 h), D1, D2, D7, and D14 for experiment 2.

### 4.4. Milk Composition

Milk composition, including fat, protein, lactose, and SNF, was analyzed at different time points. In experiment 1, measurements were conducted on D0 and D7 using the MilkoScope Expert Automatic Milk Analyzer (Etcon Analytical and Environmental Systems & Services Ltd., Lagos, Nigeria). In experiment 2, milk composition was assessed at D0, D7, and D14 at the laboratory of the Livestock Research Institute, Hsinchu Branch, Taiwan.

### 4.5. Milk Somatic Cell Counts and Cure Rate on D14

Milk SCC was measured at all sampling time points using a commercial SCC testing device (High View Innovation Co., Hsinchu, Taiwan) in accordance with the manufacturer’s instructions. In experiment 2, cure rates on D14 were categorized as either cytological or bacteriological. A cytological cure was defined as an SCC of less than 200,000 cells/mL, while a bacteriological cure was defined as the absence of detectable pathogens in milk samples. Bacteriological analysis was conducted following the protocols outlined in the Laboratory Handbook on Bovine Mastitis [45]. Briefly, 50 μL of milk was plated onto trypticase soy agar supplemented with 5% sheep blood (Dybo Enterprise Co., Ltd., Tainan, Taiwan) and incubated aerobically at 37 °C for 18–24 h. Bacterial identification was performed based on colony morphology, Gram staining, and standard biochemical assays, including the catalase and cytochrome oxidase tests.

### 4.6. Isolation of Milk Somatic Cells

The method for somatic cell separation in milk was adapted from Lee et al. (2006) [37], with modifications. Milk samples were diluted with an equal volume of phosphate-buffered saline (PBS) containing 2% fetus bovine serum (FBS) and centrifuged at 700× *g* for 20 min at 20 °C. The fat layer and supernatant were discarded, and the cell pellet was washed twice and resuspended in sterile PBS with 2% FBS.

### 4.7. Reactive Oxygen Species Level and Phagocytic Activity of Milk Somatic Cells

The levels of ROS and phagocytic activity of milk somatic cells were assessed on D0 and D2 in both experiments. The ROS levels were measured using the Invitrogen^TM^ CellROX^TM^ Deep Red Reagent (Thermo Fisher Scientific Inc.,Waltham, MA, USA), while phagocytic activity was evaluated using the Vybrant^TM^ Phagocytosis Assay Kit (Thermo Fisher Scientific Inc.,Waltham, MA, USA). In both assays, 2 × 10^5^ viable somatic cells suspended in PBS containing 2% FBS were seeded into a 96-well black plate with a clear bottom and centrifuged at 300× *g* for 7 min at room temperature to facilitate cell sedimentation. For ROS measurement, the supernatant was carefully removed, and 100 μL of 10 μM CM-H_2_DCFDA solution was added to each well, followed by incubation at 37 °C with 5% CO_2_ in the dark for 10 min. After incubation, cells were centrifuged under the same conditions, and the supernatant was replaced with 100 μL of sterile PBS containing 2% FBS. The plate was incubated again at 37 °C with 5% CO_2_ in the dark for 30 min to stabilize the cells before ROS levels were quantified using a fluorescence spectrophotometer (Synergy H1 Hybrid, BioTek, Milano, Italy). For the assessment of phagocytic activity, following centrifugation and removal of the supernatant, 100 μL of phagocytic particle solution was added to each well, and the plate was incubated at 37 °C with 5% CO_2_ in the dark for 2 h. After incubation, the particle solution was removed, and 100 μL of trypan blue solution was added for 1 min to shield non-internalized particles. Phagocytic activity was then measured using the fluorescence spectrophotometer (Synergy H1 Hybrid, BioTek, Milano, Italy).

### 4.8. Analysis of Cytokine Expression

Total ribonucleic acid (RNA) was extracted from somatic cells in experiment 2 using the TANBead Nucleic Acid Extraction 6K2 kit (Taiwan Advanced Nanotech Inc., Taiyuan, Taiwan) in combination with the Maelstrom Switch 8 automated nucleic acid extraction system (Taiwan Advanced Nanotech Inc., Taiyuan, Taiwan). Reverse transcription (RT) was performed using the PrimeScript RT Reagent Kit (Takara Bio Inc., Kyoto, Japan) according to the manufacturer’s protocol. The synthesized complementary deoxyribonucleic acid (cDNA) was stored at −20 °C until further use in a quantitative real-time polymerase chain reaction (PCR). Real-time PCR was conducted to amplify the target gene sequences and quantify messenger RNA (mRNA) levels using a StepOnePlus^TM^ Real-Time PCR System (Applied Biosystems^TM^, Waltham, MA, USA) with a SYBR Green-based detection system. The PCR reaction mixture (20 μL) contained 10 μL of PowerTrack^TM^ SYBR Green Master Mix (Applied Biosystems^TM^, Waltham, MA, USA), 500 nM of each primer, 10 ng of cDNA template, and nuclease-free water to adjust the final volume. The thermal cycling conditions were as follows: initial denaturation at 95 °C for 2 min, followed by 40 cycles of denaturation at 95 °C for 5 sec and annealing/extension at 60 °C for 30 sec. Gene expression was analyzed using the relative quantification method described by Lee et al. (2006) [37]. The expression levels of selected cytokine genes were normalized to the reference gene, bovine glyceraldehyde-3-phosphate dehydrogenase (GAPDH), at each time point and presented as relative expression ratios relative to D0. Specific primer sequences are illustrated in Table 1.

### 4.9. Statistical Analyses

All statistical analyses were conducted using Statistical Product and Service Solutions (SPSS Stat 29, IBM, Armonk, NY, USA). Given the approximately normal distribution of the data, SCC was log-transformed to SCS using the following formula:SCS = log_2_ (SCC/100,000) + 3,
where SCC is expressed in cells per milliliter, as described by Schutz et al. (1995) [23]. Data were analyzed using one-way repeated-measures analysis of variance (ANOVA), followed by Fisher’s Least Significant Difference (LSD) test for multiple comparisons. To account for individual variability, ROS levels and phagocytic activity were expressed as relative ratios with respect to D0. Comparisons of ROS levels, phagocytic activity, and relative cytokine gene expression ratios between D0 and subsequent time points were assessed using a paired-sample t-test. Statistical significance was set at *p* < 0.05, and all data are reported as mean ± standard error of the mean (SEM).

## 5. Conclusions

Intramammary infusion of SA demonstrated significant therapeutic efficacy in treating SCM in dairy cows, as evidenced by a rapid increase in SCC. The administration of SA induced an immunostimulatory response that recruited and activated immune cells, thereby enhancing pathogen clearance through enhanced natural immunity. This response led to a 54.5% bacteriological cure rate, and the sustained reduction in SCC resulted in a 58.3% cytological cure rate. The immunomodulatory effects of SA are mediated through several mechanisms, including IL-8-mediated neutrophil recruitment, IL-12-induced IFN-γ expression enhancing leukocyte activation, and IL-10 upregulation, ensuring a controlled inflammatory response. The downregulation of pro-inflammatory cytokines further supports SA’s role in facilitating termination of inflammation, contributing to sustained SCC reduction. These findings indicate that SA exerts its therapeutic effects through immune stimulation and regulation, improving innate immune responses while promoting inflammation resolution. Given its ability to modulate immune function without altering milk composition, SA offers a promising non-antibiotic alternative for SCM treatment, enhancing udder health and supporting antimicrobial stewardship in the dairy industry.

## Figures and Tables

**Figure 1 ijms-26-05515-f001:**
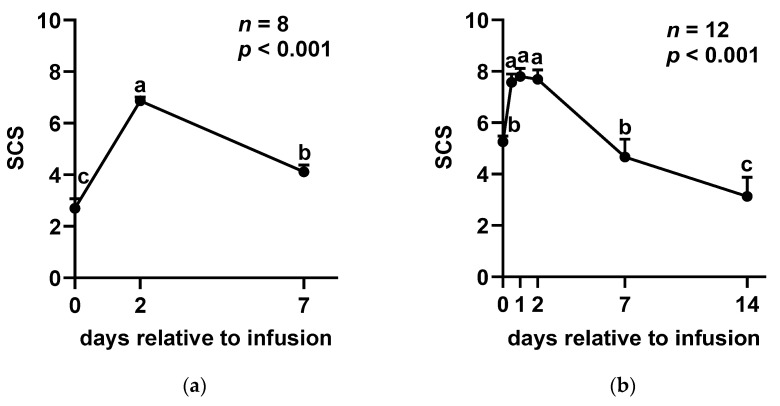
Changes in somatic cell score (SCS) over time following sodium alginate treatment in (**a**) experiment 1 (healthy quarters) and (**b**) experiment 2 (quarter with SCM). Somatic cell score, calculated as log_2_ [SCC (cells/mL)/100,000] + 3, according to Schutz et al. (1995) [23]. Each value is expressed as the mean ± SEM. a, b, c Values with different superscripts are significantly different among time points (*p* < 0.05).

**Figure 2 ijms-26-05515-f002:**
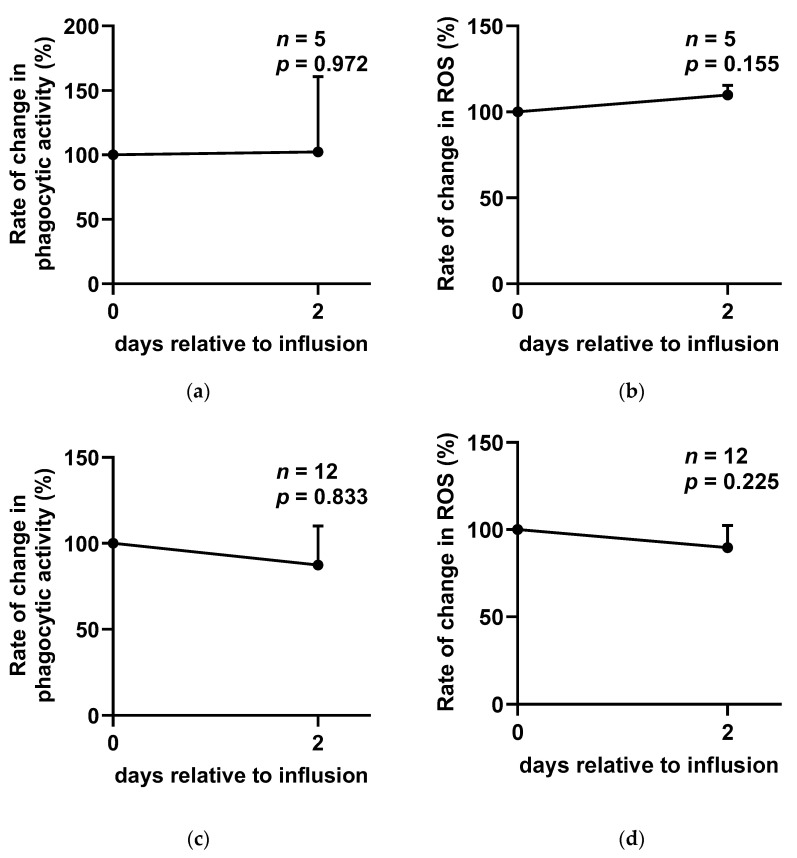
Phagocytic activity and reactive oxygen species (ROS) levels of milk somatic cells before and after sodium alginate treatment in (**a**,**b**) experiment 1 and (**c**,**d**) experiment 2. Each value is expressed as the mean ± SEM.

**Figure 3 ijms-26-05515-f003:**
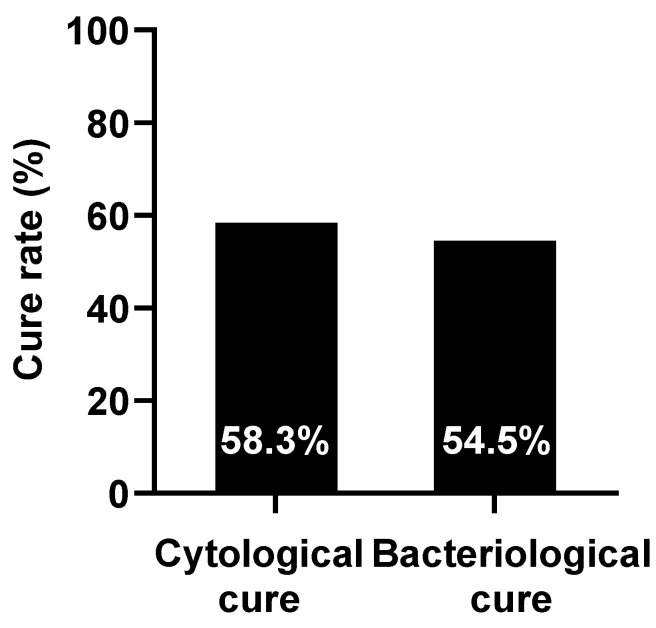
Cytological and bacteriological cure rates on D14 in experiment 2. Each value is presented as the mean (*n* = 12 and 11, respectively).

**Figure 4 ijms-26-05515-f004:**
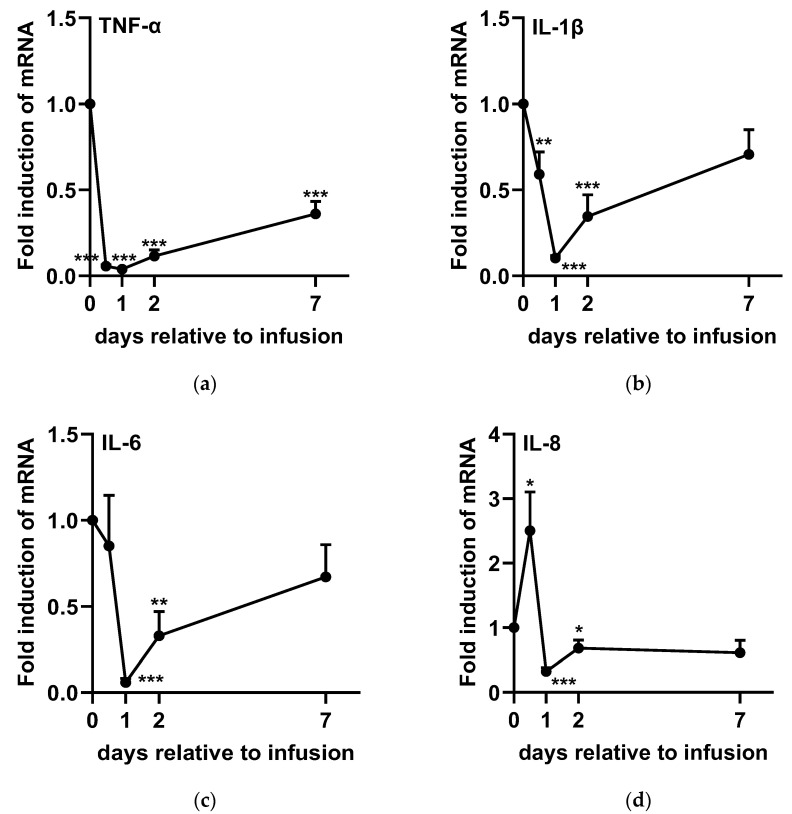
Expression levels of cytokines [(**a**) TNF-α, (**b**) IL-1β, (**c**) IL-6, and (**d**) IL-8] involved in neutrophil accumulation in bovine milk somatic cells following sodium alginate treatment in experiment 2. Each value is expressed as the mean ± SEM (*n* = 12). A *p*-value of < 0.05 was considered statistically significant when comparing D0 with each time point. Statistical significance is indicated as follows: * *p* < 0.05; ** *p* < 0.01; *** *p* < 0.001.

**Figure 5 ijms-26-05515-f005:**
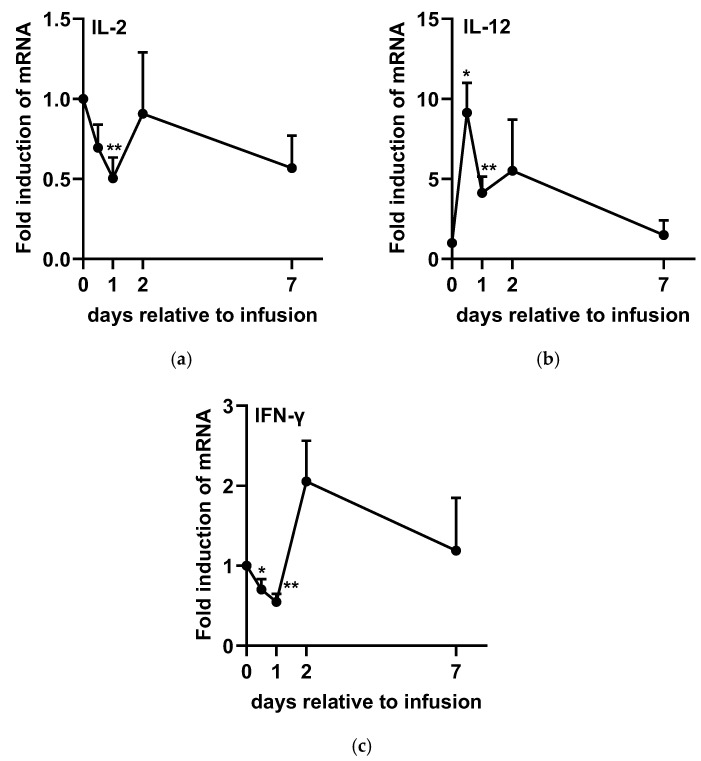
Expression levels of cytokines [(**a**) IL-2, (**b**) IL-12, and (**c**) IFN-γ] associated with neutrophil activation in bovine milk somatic cells following sodium alginate treatment in experiment 2. Each value is expressed as the mean ± SEM (*n* = 12). A *p*-value of <0.05 was considered statistically significant when comparing D0 with each time point. Statistical significance is indicated as follows: * *p* < 0.05; ** *p* < 0.01.

**Figure 6 ijms-26-05515-f006:**
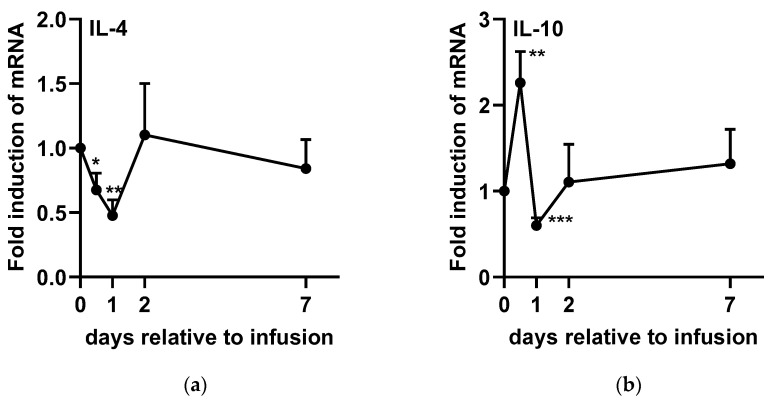
Expression levels of cytokines [(**a**) IL-4 and (**b**) IL-10] associated with inflammation resolution in bovine milk somatic cells following sodium alginate treatment in experiment 2. Each value is expressed as the mean ± SEM (*n* = 12). A *p*-value of <0.05 was considered statistically significant when comparing D0 with each time point. Statistical significance is indicated as follows: * *p* < 0.05; ** *p* < 0.01; *** *p* < 0.001.

**Table 1 ijms-26-05515-t001:** Sequences of primers for bovine GAPDH and each cytokine in real-time PCR.

Gene ^1^	Primer	Sequence (5’-3’)	Reference
GAPDH	Forward	GATTGTCAGCAATGCCTCCT	[46]
	Reverse	GGTCATAAGTCCCTCCACGA	
TNF-α	Forward	CCTGGTACGAACCCATCTA	[46]
	Reverse	ATCCCAAAGTAGACCTGCC	
IL-1β	Forward	AAAGCTTCAGGCAGGTGGTG	[46]
	Reverse	TGCGTAGGCACTGTTCCTCA	
IL-2	Forward	GAAAGTTAAAAATCCTGAGAACCTCAA	[46]
	Reverse	GCGTTAACCTTGGGCACGTA	
IL-4	Forward	AGGAGCCACACGTGCTTGA	[46]
	Reverse	TTGCCAAGCTGTTGAGATTCC	
IL-6	Forward	ATGACTTCTGCTTTCCCTACCC	[47]
	Reverse	GCTGCTTTCACACTCATCATT	
IL-8	Forward	ACACATTCCACACCTTTCCA	[48]
	Reverse	GGTTTAGGCAGACCTCGTTT	
IL-10	Forward	TTCTGCCCTGCGAAAACAA	[46]
	Reverse	TCTCTTGGAGCTCACTGAAGACTCT	
IL-12	Forward	CATCAGGGACATCATCAAACCA	[46]
	Reverse	CCTCCACCTGCCGAGAATT	
IFN-γ	Forward	GTAGCCCTGTGCCTGATTTC	[46]
	Reverse	CACATTGTCCCTTCCCAGAG	

^1^ GAPDH, Glyceraldehyde-3-phosphate dehydrogenase; TNF-α, Tumor necrosis factor-alpha; IL, interleukin; IFN-γ, interferon gamma.

## Data Availability

The original contributions presented in this study are included in the article, and further inquiries are available from the corresponding authors.

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
