# Peer review of "Effects of Sodium Alginate Infusion on Intramammary Immunity Against Subclinical Mastitis in Dairy Cows"

_ijms, 2025, doi:10.3390/ijms26125515_

Round 1

Reviewer 1 Report

Comments and Suggestions for Authors

Article ID: ijms-3616820

“Effects of sodium alginate infusion on the intramammary immunity against subclinical mastitis in dairy cows”

General remarks

The manuscript is interesting, in particular the use of an alternative compound, instead of antibiotics, to improve subclinical mastitis problematic in dairy cows. All sections are readable and written in a good English. Although it is not mandatory, as the journal do not have strict formatting requirements, I suggest putting the materials and methods section between the introduction and the results, to improve the fluency and make easier the reading of your work. Moreover, the authors should check the journal instructions about how to write an article, as conclusions is one of the required sections. I have some concern about the absence of results and discussion of some parameters in the Experiment 1, making impossible seeing the effect of sodium alginate on healthy cows and also the comparison of these parameters between the healthy dairy cows and the ones with subclinical mastitis.

My additional specific comments are listed below, point by point. I hope my suggestions will be helpful in improving the quality of the manuscript.

Specific comments

Abstract

ok

  1. Introduction

ok

  1. Results

Lines 88, 93, and so on: please be more specific when you refer to the figure adding also the respective letters.

Line 107: p should be in italic. Please correct.

From line 103 to 116: Why did you only present data for Experiment 2? If the data from Experiment 1 were not statistically significant, this should be stated somewhere.

Figure 1: I suggest ordering the significance letters from highest to lowest value.

Figure 5-b: the significance assignment seems incorrect. How can the difference between D0 and D1 be higher than the difference between D0 and D0.5? Moreover, are you sure that there is no difference between D0 and D2? Please check and correct or explain it.

Figure 6-b: as before, I recommend checking and, if necessary, correcting the significance assignment.

  1. Discussion

As for the results section, some parameters are commented only for Experiment 2. Could you explain why?

  1. Materials and Methods

ok

Conclusions

Please add a proper conclusion to your work.

Author Response

Comment 1: I suggest putting the materials and methods section between the introduction and the results to improve fluency and readability.
Response 1: Thank you for your suggestion. We have revised the manuscript’s structure accordingly and moved “Materials and Methods” before the “Results”.

Comment 2: Results, Lines 88, 93, and so on: please be more specific when you refer to the figure adding also the respective letters.
Response 2: We have added the respective figure letters and clarified the figure references throughout the manuscript (Lines 202–229).

Comment 3: Results, Line 107: p should be in italic. Please correct.
Response 3: Thank you for pointing out the error. We have corrected the formatting of p values accordingly (Line 220).

Comment 4: Results, From line 103 to 116: Why did you only present data for Experiment 2? If the data from Experiment 1 were not statistically significant, this should be stated somewhere.
Response 4: Thank you for your insightful comment. Experiment 1 was conducted as preliminary study to evaluate the response to intramammary infusion of sodium alginate in healthy dairy cows. Results demonstrated that intramammary infusion of sodium alginate is able to increase SCC, presumably due to increase of migrated neutrophils, but not phagocytosis and ROS. Moreover, intramammary infusion of sodium alginate does not induce prolonged increase of SCC as indicated by gradually declined SCC and unaltered milk composition (data not shown). Therefore, cytokine expressions of SCC were not analyzed in Experiment 1. In contrast, Experiment 2 was carried out to access the therapeutic efficacy and immunological mechanisms elicited by intramammary infusion of sodium alginate in cows with subclinical mastitis, in which cytokine gene expressions were analyzed.

Comment 5: Results, Figure 1: I suggest ordering the significance letters from highest to lowest value.
Response 5: We appreciate your suggestion. We have revised the lettering in Figure 1a and 1b to follow the order from highest to lowest values (Page 6).

Comment 6: Results, Figure 5-b: the significance assignment seems incorrect. How can the difference between D0 and D1 be higher than the difference between D0 and D0.5? Moreover, are you sure that there is no difference between D0 and D2? Please check and correct or explain it.
Response 6: Thank you for highlighting this issue. We have corrected the significance labeling in Figure 5b and rechecked all values (Page 9). The lack of significance between D0 and D2 is due to the high standard deviation within the D2 group.

Comment 7: Results, Figure 6-b: as before, I recommend checking and, if necessary, correcting the significance assignment.
Response 7: We have double-checked the statistical analysis for Figure 6b. The significance assignment is correct, and no changes were necessary (Page 9).

Comment 8: Discussion, As for the results section, some parameters are commented only for Experiment 2. Could you explain why?
Response 8: As explained in the response to Comment 4, analysis of cytokine gene expressions was not conduced in Experiment 1, so that we did not have results to discuss the profile of cytokine gene expression in response to intramammary infusion of sodium alginate in healthy cows. A sentence “However, analysis of cytokine gene expressions was only conducted in Experiment 2 to elucidate more detailed immunomodulating properties of SA in SCM cows.” was added (line 302-304) to clarify this.  

Comment 9: Conclusions, Please add a proper conclusion to your work.
Response 9: Thank you for your suggestion. The conclusion, previously embedded in the “Discussion” section, has now been stated separately in "Conclusion" section as suggested (Pages 11–12; Lines 353–369).

Reviewer 2 Report

Comments and Suggestions for Authors

General review

The topic and the quality of this paper is suitable for the International Journal of Molecular Sciences Journal. The bovine mastitis (clinical and subclinical) is a really great problem in the dairy herds worldwide. Subclinical mastitis has a great effect on the economic results of dairy herds. Decreasing the prevalence of subclinical mastitis is a great important task in the dairy sector.

Detailed review

Abstract

line 23: please add the before infusion word “intramammary” phrase!

Introduction

line 41: please add clinical and subclinical phrases into the “The global prevalence of mastitis …”!

This section contains the most important (and adequate) references which are connected to the topic of the manuscript.

Results

Figure 1.: please add the unit into the Figure! Moreover, does SCC scale normal or converted values (pl log)?

Figure 2.: maybe better if presented these results in a Table! I think a Table is enough!

Discussion

This section is well-prepared.

Materials and Methods

4.1: please add the mean values of SCC in the results section!!!

4.5.: how to detect the udder pathogens (I think, the authors thought udder pathogens not other pathogens…)? But the authors used a reference, but in this relation (and importance) need more info about detection of udder pathogens! Please explain it!

Conclusions

Conclusions are appropriate.

Author Response

Comment 1: Abstract, line 23: please add the before infusion word “intramammary” phrase!
Response 1: Thank you for the suggestion. We have revised the sentence accordingly (Page 1, Line 23).

Comment 2: Introduction, line 41: please add clinical and subclinical phrases into the “The global prevalence of mastitis …”!
Response 2: This sentence has been revised as suggested (Page 1, Line 41).

Comment 3: Results, Figure 1.: please add the unit into the Figure! Moreover, does SCC scale normal or converted values (pl log)?
Response 3: The somatic cell count (SCC) was converted into somatic cell score (SCS). We have added a description of the conversion formula (Page 5, Lines 189–192; Page 6, Lines 233–234). Since the values are numerically transformed, no unit is presented.

Comment 4: Results, Figure 2.: maybe better if presented these results in a Table! I think a Table is enough!
Response 4: Thank you for the suggestion. Although no significant differences were observed in phagocytic activity and ROS levels in both experiment 1 and 2, we thought that a graphical representation can better convey the overall trends between D0 and D2 to readers. Therefore, please allow us to maintain these data in a figure format (Page 7; Figure 2).

Comment 5: Materials and Methods, 4.1: please add the mean values of SCC in the results section!!!
Response 5: We have added the mean SCC values as requested (Page 2–3; Lines 92, 93, 95, and 96). The average SCC was 72,750 ± 17,670 cells/mL in Experiment 1, and 541,417 ± 81,708 cells/mL in Experiment 2.

Comment 6: Materials and Methods, 4.5.: how to detect the udder pathogens (I think, the authors thought udder pathogens not other pathogens…)? But the authors used a reference, but in this relation (and importance) need more info about detection of udder pathogens! Please explain it!
Response 6: Thank you for your suggestion. We have expanded the description as the followings (Page 3; Lines 125–129):
“Briefly, 50  μL of milk was plated onto trypticase soy agar supplemented with 5% sheep blood (Dybo Enterprise Co., Ltd., Tainan, Taiwan) and incubated aerobically at 37 °C for 18–24 hours. Bacterial identification was based on colony morphology, Gram staining, and standard biochemical assays, including catalase and cytochrome oxidase tests.”
